# Metabolic disorders and the risk of head and neck cancer: a protocol for a systematic review and meta-analysis

Alexander Gormley [1,2] Charlotte Richards [3] Francesca Spiga,[4,5] Emily Gray,[2] Joanna Hooper,[6] Barry Main [1,4] Emma E Vincent,[5,7] Rebecca Richmond,[4,5] Julian Higgins,[4,5] Mark Gormley[1,5]

AG and CR are joint first authors.

**Correspondence to**
Mr Mark Gormley;
mark.gormley@bristol.ac.uk

## ABSTRACT

**Introduction** Head and neck cancer squamous cell carcinoma (HNSCC) is the sixth most common cancer internationally. Established risk factors include smoking, alcohol and presence of human papillomavirus (HPV). The incidence rate of new disease continues to rise, despite falls in alcohol consumption and a reduction in smoking, the rising rates are unlikely to be solely attributed to HPV status alone. Obesity and its associated conditions such as type 2 diabetes (T2D) are implicated in the risk and progression of a variety of cancers, but there is paucity of evidence regarding its role in HNSCC.

**Methods and analysis** A systematic review of cohort studies, reporting a risk of incident HNSCC, will be included. A systematic search strategy has been developed, multiple databases will be searched from January 1966 to November 2021, including Cochrane Library, OVID SP versions of Medline and EMBASE. The primary outcome will be incident HNSCC based on exposures of T2D, obesity, dyslipidaemia and hypertension as defined by the WHO. A combined risk effect across studies will be calculated using meta-analysis, although depending on the heterogeneity in study design, exposure and outcome reporting this may not be possible.

**Ethics and dissemination** No ethical approval is required for this systematic review. The review will be published in a relevant peer-review journal and findings will be presented at scientific meetings in both poster and oral presentation form.

**PROSPERO registration number details** This study has been registered with the International Prospective Register of Systematic Reviews (PROSPERO) with study registration number CRD42021250520. This protocol has been developed in accordance with the Preferred Reporting Items for Systematic Review and Meta-Analysis Protocols guidance statement.

## INTRODUCTION

Head and neck cancer squamous cell carcinoma (HNSCC), which includes cancer of the oral cavity, oropharynx, hypopharynx and larynx, is the world's sixth most common cancer, with the highest incidence in men and those over 70 years old.[1] Established risk factors include smoking, alcohol and the human papillomavirus (HPV), which has

---

## STRENGTHS AND LIMITATIONS OF THIS STUDY

⇒ This systematic review will be the first to comprehensively review the literature and provides a meta-analysis of the effect of metabolic disorders and the risk of incident head and neck cancer squamous cell carcinoma (HNSCC).

⇒ The publication of this protocol provides a clear representation of the methods used in this review for transparency and to prevent future duplication.

⇒ This systematic review will be one of the first to use the Risk Of Bias In Non-randomised Studies—of Exposures tool.

⇒ Despite the metabolic disorders being described as separate entities, the authors recognise the potential for disease processes to be related.

⇒ Among cohort studies, there is significant variability in terms of length of follow-up for the outcome of interest, HNSCC.

---

mainly been linked to oropharyngeal cancer.[2] Despite the significant reduction in smoking,[3] and a fall in alcohol consumption,[4] in the UK over the previous two decades, incidence rates of head and neck cancers have continued to rise by around a third.[1] Given this potentially changing aetiology,[2] further exploration of less well known risk factors is warranted to target prevention and identify patients with early disease. People living with obesity and associated conditions, such as type 2 diabetes (T2D), have an increased risk of developing certain cancers (eg, liver and pancreas), but the epidemiological evidence surrounding HNSCC is not conclusive.[5 6] Recent studies have demonstrated elevated glucose levels before or at the time of diagnosis, which may play a role in the pathogenesis and progression of cancer.[7] Carcinogenesis in HNSCC is driven by diverse signalling pathways; however, little is known about the role of metabolism,[8] despite metabolic reprogramming being a recognised hallmark of cancer.[9]

T2D is a common condition characterised by insulin resistance and hyperglycaemia, which results in whole-body metabolic dysregulation.[10] In 2015, the estimated worldwide diabetes prevalence for adults was predicted to rise to 642 million by 2040.[11] In addition, public health strategies have not successfully addressed the current 'obesity pandemic'.[12–16] One large pooled analysis demonstrated a weak positive association between T2D and HNSCC (OR 1.09; 95% CI: 0.95 to 1.24), which was stronger among those who never smoked cigarettes (OR 1.59; 95% CI: 1.22 to 2.07).[6] While obesity is a potent risk factor for T2D, the relationship between body mass index (BMI) and HNSCC is not straightforward, with both positive and inverse associations observed.[17] Another study found that a positive association with BMI was only observed in never smokers.[18] In the largest study to date, waist circumference and waist-to-hip ratio were positively associated with higher HNSCC risk, regardless of smoking status.[18] Metabolic syndrome has been described as a clustering of disorders including obesity, hypertension, hyperglycaemia and dyslipidaemia. A moderate inverse association has been observed between metabolic syndrome and HNSCC (OR 0.81; 95% CI: 0.78 to 0.85), but again these results were modified by tobacco use.[19] Treatment for these conditions using medications such as statins,[20] or metformin,[21] may also reduce the risk of developing HNSCC. Given the growing pandemic of metabolic disorders, understanding how these alterations affect the risk of carcinogenesis may identify those who are high risk. This could drive targeted prevention, earlier detection and perhaps the identification or repurposing of drug targets for treatment in HNSCC.[22]

## Rationale

The complex metabolic changes associated with T2D, obesity, dyslipidaemia and hypertension may alter the risk of certain cancers, but the evidence for head and neck cancer is inconclusive. This study aims to identify, collate and synthesise all relevant studies including adult participants, to determine whether the risk of developing incident HNSCC is influenced by T2D, obesity, dyslipidaemia and hypertension, using relevant effect measures (eg, OR or risk ratios (RR)). Where available, we will further stratify by subsite (ie, oral cavity, oropharynx, hypopharynx and larynx), as well as by HPV status.

## METHODS

### Research question

Do metabolic disorders affect the risk of developing head and neck cancer?

### Study design

Randomised controlled trials are informative for assessing causal effects; however, due to the nature of the exposures in this study, inclusion of this design is not feasible. Therefore, we will focus on observational (eg, cohort studies) reporting the risk of metabolic disorders on incident HNSCC.

### Eligibility criteria

Population: participants over 18 years old, of either sex, from any ethnic background.

Exposures (1) T2D, (2) obesity, (3) dyslipidaemia and (4) hypertension. These will be collectively described as 'metabolic disorders'.

Comparison: participants who have not been diagnosed with the afore-mentioned metabolic disorders.

Outcome: incident HNSCC.

The study will report in line with the Preferred Reporting Items for Systematic Reviews and Meta-Analyses (PRISMA) statement and has been pre-registered on International Prospective Register of Systematic Reviews (PROSPERO) in 2021.

### Search strategy

A systematic search strategy (online supplemental table 1) has been formulated by clinicians, scientific researchers and a specialist librarian for surgery, head and neck and medicine. Medical Subject Headings and keywords will be iteratively combined with the Boolean operators AND, OR and NOT. Multiple databases will be searched from January 1966 to November 2021, including OVID SP versions of Medline and EMBASE, using the formulated search strategy (online supplemental table 1). Preprint servers including medRxiv and bioRxiv will also be searched. In addition, the following electronic bibliographic databases will be searched: Cochrane Library, EThOS, Google Scholar, Open Grey and ClinicalTrials.gov, to identify articles from the grey literature and conference proceedings using a modified search strategy (online supplemental table 2). References extracted from the full-length articles will be reviewed to identify other publications of interest. Duplicate articles will be removed using Covidence.[23]

### Data management

The results from searches will be imported into Covidence software, to improve reference management and workflow. This will also populate a PRISMA flow diagram.[24]

### Data screening and extraction

All titles and abstracts will be screened by two authors AG and CR, with conflicts resolved by MG. Screening records and decisions will be kept in Covidence. Data will be extracted from titles that meet the inclusion criteria (table 1). The data extraction form (Online supplemental table 3) will be piloted using five included titles with AG, CR and MG independently completing data extraction. If amendments are necessary, they will be performed at this point, with consensus approval required prior to final use of the form. Subsequent to this, CR and MG will double data extract independently for each title, any disagreements will be discussed among AG, CR and MG with consensus approval required.

**Table 1** Study selection criteria

| Inclusion criteria | Exclusion criteria |
|---|---|
| All studies published from January 1966 in the English language. | The study is not based on incident head and neck squamous cell carcinoma, or contains only prevalent data, meaning a temporal relationship cannot be inferred. |
| Participants humans>18 years old, of either sex and any ethnic background. | Studies focused on cancer survival or progression. |
| Observational (eg, cohort studies) reporting the risk of incident head and neck squamous cell carcinoma. | Study designs such as case reports or case series. In addition, cross-sectional, case–control and narrative review studies will be excluded due to the inability to infer a temporal relationship. |
| Studies must report an OR or risk ratio, or data which will allow these to be calculated. | Human studies only, no in vivo animal or in vitro cell line studies. |
| Exposures: type 2 diabetes, obesity, dyslipidemia and hypertension with definitions as described in the study protocol. | Studies of head and neck epithelial dysplasia, potentially malignant disease or carcinoma in situ. |
| An outcome of head and neck cancer diagnosis which may be human papilloma virus (HPV) positive or negative. High risk types HPV16, 18, 31 and 33 only will be included. | Squamous cell carcinoma of other sites, for example, nasopharynx, salivary gland, oesophagus, skin or lung due to differing aetiology, histological subtypes and risk factors, for example, Epstein-Barr virus. |

## Exposure definitions

1. The WHO defines T2D as a chronic disease that occurs when the body cannot effectively use the insulin it produces and is largely the result of excess body weight and physical inactivity. A diagnosis of T2D was defined as symptoms such as polyuria or polydipsia, plus[25 26]:
   - A random blood plasma glucose concentration≥11.1 mmol/L.
   - A fasting plasma glucose concentration≥7.0 mmol/L (whole blood≥6.1 mmol/L).
   - Two-hour plasma glucose concentration≥11.1 mmol/L, two hours after 75 g anhydrous glucose in an oral glucose tolerance test.
   - Glycated haemoglobin (HbA1c) 6.5% or more (48 mmol/mol and above).

   Relevant International Classification of Disease (ICD) codes used to define T2D will also be acceptable.

2. The WHO definition of BMI of 30 or above must have been used to define obesity.[27] BMI if calculated from self-reported height and weight or taken from medical records or measured at baseline will be acceptable.

3. Dyslipidaemia was classified as serum total cholesterol, low-density lipoprotein cholesterol, triglycerides, apolipoprotein B or lipoprotein(a) concentrations above the 90th percentile, or high-density lipoprotein cholesterol or apolipoprotein concentrations below the 10th percentile for the general population.[28] Relevant ICD codes used to define dyslipidaemia will also be acceptable.

4. The WHO definition of hypertension as[29]:
   - A systolic blood pressure reading of ≥140 mm Hg.
   - A diastolic blood pressure readings on both days of ≥90 mm Hg, must have been used.

   Relevant ICD codes used to define hypertension will also be acceptable.

## Outcome definitions

All diagnoses of incident HNSCC included in this study should have been confirmed by histology by a trained pathologist. Acceptable cancer diagnosis may be reported using ICD codes.[30] Cancer cases for the disease of interest have the following ICD-10 codes: oral cavity (C02.0–C02.9, C03.0–C03.9, C04.0–C04.9, C05.0–C06.9) oropharynx (C01.9, C02.4, C09.0–C10.9), hypopharynx (C13.0–C13.9), overlapping (C14 and combination of other sites) and 25 cases with unknown ICD code (other). Older versions of ICD code, for example, ICD-8 and ICD-9 were also accepted. With respect to HPV diagnosis, only high-risk subtypes HPV16, 18 31, and 33 will be included. These are should be detected using HPV-DNA via PCR and/or in situ hybridisation examination, usually used in combination with immunohistochemistry for p16.

## Data analysis and synthesis

The analysis and synthesis process will be performed in two phases. The first phase consists of estimating the effect of each metabolic disorder (T2D, obesity, dyslipidaemia and hypertension) on incident HNSCC separately for each included study. Effects should be reported as HR, standardised incidence ratio, OR or RR, with 95% CIs or in a format that will enable us to calculate these.

The second phase will focus on estimating the combined effect across studies using meta-analysis, using metan software in Stata (Stata Statistical Software: Release V.16. College Station, Texas, USA: StataCorp LLC). Summary effects with their corresponding 95% CIs will be derived with the method of DerSimonian and Laird.[31] This will be performed separately for each metabolic disorder. The analysis will yield information about the heterogeneity of effects across studies, using Cochran's Q test and Higgins' $I^2$ statistic. Due to the heterogeneity in study

design, exposure definition and outcome reporting in the included studies, meta-analysis may not be reasonable. If appropriate, to further explore the sources of heterogeneity if present, meta-regression analyses will be performed according to age, geographic location, year of publication, methods of exposure assessment, risk of bias score, length of follow-up, the number of cases and adjustment for confounding factors including smoking and alcohol use. Small study effects, which may reflect publication bias, study quality or other sources of heterogeneity, will be assessed using funnel plots and Egger's regression asymmetry tests.

## Subgroup analyses

Where possible, we will stratify by oral, oropharyngeal, hypopharyngeal and laryngeal subsite and by HPV status as post-hoc analyses, to determine if effects seen are specific to a particular anatomical region or associated with specific tumour types.

## Risk of bias in studies

Two authors will independently extract the relevant risk factor information during data extraction using a preliminary version of the Risk of Bias in Non-randomised Studies—of Exposures (ROBINS-E) tool (https://www.bristol.ac.uk/population-health-sciences/centres/cresyda/barr/riskofbias/robins-e/).[32] The ROBINS-E tool aims to assess the result of interest from an observational study for risk of bias, where that study is designed to assess the effect of exposure on outcome.[32] Disagreements will be recorded and discussed with a third author for resolution. This assessment will be used in evaluating the strength of evidence from the included studies.

## Amendments

Any amendments to the initial protocol will be included in an update to the PROSPERO record, including information on amendment type, reasoning and a timestamp. Deviation from this protocol will be described in the full systematic review paper.

## DISSEMINATION

This systematic review will be published in a relevant peer-reviewed journal and findings from this work may be presented at scientific meetings (both poster and oral presentations).

## Author affiliations
[1] Bristol Dental School, Faculty of Health Sciences, University of Bristol, Bristol, UK
[2] Bristol Dental Hospital, University Hospitals Bristol and Weston NHS Foundation Trust, Bristol, UK
[3] School of Dentistry, Cardiff University, Cardiff, UK
[4] Population Health Sciences, Bristol Medical School, Faculty of Health Sciences, University of Bristol, Bristol, UK
[5] MRC Integrative Epidemiology Unit, Population Health Sciences, Bristol Medical School, Faculty of Health Sciences, University of Bristol, Bristol, UK
[6] Library and Information Services, University Hospitals Bristol and Weston NHS Foundation Trust, Bristol, UK
[7] School of Cellular and Molecular Medicine, Faculty of Life Sciences, University of Bristol, Bristol, UK

**Contributors** AG and CR jointly contributed to the development of the protocol and the drafting, writing and editing of this manuscript. FS contributed to the development of this study and editing of the manuscript. JH contributed to the development of the search strategy. BM, EG, EEV, RR and JH contributed to the development of this study. MG was responsible for conceptualising this study, drafting and editing this manuscript. All authors contributed to the development of the search strategy. All authors have approved and contributed to the final written manuscript.

**Funding** AG is a National Institute for Health Research (NIHR) academic clinical fellow (no grant number). FS was supported by a Cancer Research UK (C18281/A29019) programme grant (the Integrative Cancer Epidemiology Programme). EEV is supported by Diabetes UK (17/0005587). EEV is also supported by the World Cancer Research Fund (WCRF UK), as part of the World Cancer Research Fund International grant programme (IIG_2019_2009). RR. is a de Pass VC research fellow at the University of Bristol (no grant number). JH supported by the NIHR Biomedical Research Centre at University Hospitals Bristol and Weston NHS Foundation Trust and the University of Bristol (no grant number), the NIHR Applied Research Collaboration West at University Hospitals Bristol and Weston NHS Foundation Trust (no grant number) and is an NIHR Senior Investigator (no grant number). MG is currently supported by a Wellcome Trust GW4-Clinical Academic Training PhD Fellowship. This research was funded in part, by the Wellcome Trust (grant number 220530/Z/20/Z). For the purpose of open access, the author has applied a CC BY public copyright licence to any author accepted manuscript version arising from this submission. The views expressed in this publication are those of the author(s) and not necessarily those of Wellcome, NIHR, the NHS or Department of Health.

**Competing interests** None declared.

**Patient and public involvement** Patients and/or the public were not involved in the design, or conduct, or reporting, or dissemination plans of this research.

**Patient consent for publication** Not applicable.

**Provenance and peer review** Not commissioned; externally peer reviewed.

**ORCID iDs**
Alexander Gormley http://orcid.org/0000-0002-2628-9928
Charlotte Richards http://orcid.org/0000-0002-4624-1155
Barry Main http://orcid.org/0000-0003-0622-805X

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
