## [Reviewer comments · BMJ Open]

ARTICLE DETAILS

TITLE (PROVISIONAL)	Metabolic disorders and the risk of head and neck cancer: a protocol for a systematic review and meta-analysis
AUTHORS	Gormley, Alexander; Richards, Charlotte; Spiga, Francesca; Gray, Emily; Hooper, Joanna; Main, Barry; Vincent, Emma; Richmond, Rebecca; Higgins, Julian; Gormley, Mark

VERSION 1 – REVIEW

REVIEWER	Richard Shaw Institute of Translational Medicine, University of Liverpool,
REVIEW RETURNED	29-Nov-2021

GENERAL COMMENTS	Risk factors for HNSCC, as with many other cancers include advancing age, so the increasing longevity of western populations is one significant factors in increasing incidence perhaps overlooked here. The risk factors mentioned were only HPV, smoking and alcohol.... And of course there will be a confounding effect of DM and age. The authors need to be careful to avoid confirmation bias, in seeking a relationship between metabolic disorders (DM) and malignancy, for example: Increased incidence of DM related to pancreas cancer are likely a direct effect of the tumour on the pancreas itself (ie. not a metabolic effect per se). One aspect of the correlation between BMI and HNSCC not mentioned is the direct effect of primary tumours causing dysphagia and malnutrition (ie. not a metabolic effect per se)
--

REVIEWER	Francesc Xavier Aviles-Jurado Hospital Clinic de Barcelona, Otorhinolaryngology
REVIEW RETURNED	16-Jan-2022

GENERAL COMMENTS	First of all, I would like to congratulate the authors for making this attempt to systematise the study of the relationship between these pathologies and head and neck cancer. Those of us who work in head and neck pathology know that there are many dark areas and metabolic disorders can be one of them, as has been demonstrated in other types of tumours. I find the methodology followed for this systematisation impeccable, it is the great strength of the work, however I have serious discrepancies in certain aspects, which do not really agree with the usual clinic of these patients. The aspect of tumour locations is clearly insufficient, as they do not
--

	take into account the hypopharynx (although they later include it as a diagnostic code), even with the enormous volume that it represents. Nor do I understand why they do not assess the nasopharynx or why they do not take into account the ebstein barr virus. In this way, part of the oncological pathology of the head and neck is being ignored. Regarding HPV, they seem to consider diagnosis by PCR, when we are currently accepting diagnosis by p16 immunohistochemistry, due to the fact that many centres in many countries do not have molecular biology laboratories available. The key words do not seem to me to be representative The best thing for me is the attempt to protocolise a way of carrying out a meta-analysis, although it does not seem to me to be a new contribution, I value it as an important aspect. In short, it seems to me to be a good methodological study, with clinical discrepancies and I am not sure about the number of citations it can contribute to the journal.
--	--

VERSION 1 – AUTHOR RESPONSE

Reviewer: 1

Dr Richard Shaw, Institute of Translational Medicine, University of Liverpool

Reviewer comments to the author:

Risk factors for HNSCC, as with many other cancers include advancing age, so the increasing longevity of western populations is one significant factors in increasing incidence perhaps overlooked here. The risk factors mentioned were only HPV, smoking and alcohol.... And of course there will be a confounding effect of DM and age.

Thank you for this comment. We acknowledge the confounding effect of age (influencing both our metabolic disorder (exposures), and head and neck cancer (outcomes)). All studies included in our review will be comprehensively assessed for risk of bias and confounding using the Risk Of Bias In Non-randomized Studies of Exposures (ROBINS-E) tool (<https://www.bristol.ac.uk/population-health-sciences/centres/cresyda/barr/riskofbias/robins-e/>). Methods for dealing with confounding in each study will be evaluated, and adjustment for any confounders will be compared to a list of pre-specified known confounders including age.

If appropriate, meta-regression will be carried out to study possible sources of heterogeneity and to determine if specific metabolic disorders are related to age. The protocol has now been updated to reflect this:

The second phase will focus on estimating the combined effect across studies using meta-analysis, using metan software in Stata (Stata Statistical Software: Release 16. College Station, TX: StataCorp LLC). Summary effects with their corresponding 95% CIs will be derived with the method of DerSimonian and Laird.¹ This will be performed separately for each metabolic disorder. The analysis will yield information about the heterogeneity of effects across studies, using Cochran's Q test and Higgins' I² statistic. Due to the heterogeneity in study design, exposure definition and outcome reporting in the included studies, meta-analysis may not be reasonable. If appropriate, to further explore the sources of heterogeneity if present, meta-regression analyses will be performed according to age, geographic location, year of publication, methods of exposure assessment, risk of bias score, length of follow-up, the number of cases and adjustment for confounding factors including smoking and alcohol use. Small study effects, which may reflect publication bias, study quality or other sources of heterogeneity, will be assessed using funnel plots and Egger's regression asymmetry tests. (Lines 225-237, page 12).

The authors need to be careful to avoid confirmation bias, in seeking a relationship between metabolic disorders (DM) and malignancy, for example: Increased incidence of DM related to pancreas cancer are likely a direct effect of the tumour on the pancreas itself (i.e., not a metabolic effect per se).

We acknowledge the reviewer's concerns here, however, the aim of this paper is to perform an objective systematic review and meta-analysis to explore potential associations between metabolic disorders and head and neck cancer risk. Therefore, it could be we find there is an increased risk, decreased risk or null effect of metabolic disorders on head and neck cancer, depending on the exposure. The purpose of this pre-published protocol is to ensure we avoid confirmation bias, taking a systematic approach to finding and collecting data, as well as analysing and reporting in the manner we stated we would. This includes having a clear, focussed question, comprehensive search strategy,

pre-defined eligibility criteria, reproducible methodology, risk of bias assessment and transparent reporting.

Regarding the issue of reverse causality (i.e., in the described case of pancreatic cancer resulting in metabolic changes), this is usually observed in cases of short duration (<2 years) or new-onset diabetes mellitus.² This direct biological plausibility is less obvious in head and neck cancer (compared to pancreatic), however, those diagnosed with diabetes may make other changes in lifestyle which reduce cancer risk. While this is a potential limitation of observational studies, we included only studies which report on head and neck cancer incidence e.g., cohort studies, and excluded studies presenting prevalent data, whereby a temporal relationship cannot be inferred. Other designs including Mendelian randomization (MR) which aim to overcome issues with unmeasured confounding and reverse causality will also be considered eligible in our review. MR uses genetic variation (with single nucleotide polymorphisms randomly allocated at birth taken as proxies for an exposure), to investigate the causal relationships between potentially modifiable risk factors and health outcomes in observational data.

See **Table 1** (line 140) for justification on inclusion and exclusion criteria.

One aspect of the correlation between BMI and HNSCC not mentioned is the direct effect of primary tumours causing dysphagia and malnutrition (i.e., not a metabolic effect per se).

This is another good example of potential reverse causality, a limitation of epidemiological studies which we have attempted to mitigate by only including studies which report on head and neck cancer incidence as described above. Therefore, all participant data analysed in this review should have confirmed obesity at baseline (i.e., prior to their diagnosis of head and neck cancer), making the possibility for reverse cause less likely (but not impossible). Again, all studies will be assessed for risk of bias with respect to their exposure and outcome measurements using the ROBINS-E tool.

See **Table 1** (line 140) for justification on inclusion and exclusion criteria.

Reviewer: 2

Dr Francesc Xavier Aviles-Jurado, Hospital Clinic de Barcelona

Reviewer comments to the author:

First of all, I would like to congratulate the authors for making this attempt to systematise the study of the relationship between these pathologies and head and neck cancer. Those of us who work in head and neck pathology know that there are many dark areas and metabolic disorders can be one of them, as has been demonstrated in other types of tumours. I find the methodology followed for this systematisation impeccable, it is the great strength of the work, however I have serious discrepancies in certain aspects, which do not really agree with the usual clinic of these patients.

The aspect of tumour locations is clearly insufficient, as they do not take into account the hypopharynx (although they later include it as a diagnostic code), even with the enormous volume that it represents.

Thank you for this comment. We agree and will plan to include and stratify by hypopharyngeal subsite, where available. This has now been changed in our protocol as below:

Where available, we will further stratify by subsite (i.e., oral cavity, oropharynx, hypopharynx and larynx), as well as by HPV status. (Lines 114-115, page 6)

Nor do I understand why they do not assess the nasopharynx or why they do not take into account the Epstein-Barr virus. In this way, part of the oncological pathology of the head and neck is being ignored.

Non-keratinising nasopharyngeal carcinomas (NPCs) are almost always Epstein-Barr virus (EBV) positive. As non-keratinising is the most common histological subtype (>95 % of all NPCs in high incidence regions), there is a clear difference in aetiology in comparison to the other head and neck

cancer subsites.^{3,4} As per our protocol, we have also excluded salivary gland, oesophageal, skin or lung cancer due to differing aetiology, histological subtypes and risk factors to ensure better generalisability and interpretation of our results.

See **Table 1** (line 140) for justification on exclusion criteria.

Regarding HPV, they seem to consider diagnosis by PCR, when we are currently accepting diagnosis by p16 immunohistochemistry, due to the fact that many centres in many countries do not have molecular biology laboratories available.

I think there may be a misunderstanding here as we stated both PCR and p16 immunohistochemistry would be acceptable as methods of HPV detection as per our protocol. We have now re-worded this point in our protocol to ensure clarity:

With respect to HPV diagnosis, only high risk subtypes HPV16, 18, 31, and 33 will be included. These can be detected using HPV-DNA via polymerase chain reaction (PCR) and/or in situ hybridization (ISH) examination, usually used in combination with immunohistochemistry for p16. (Lines 211-215, page 11)

The key words do not seem to me to be representative.

All key words and search terms were MeSH (Medical Subject Headings) to ensure we capture all relevant publications. In addition to the previous key terms we have added: "Hypertension < CARDIOLOGY", "Lipid disorders < DIABETES & ENDOCRINOLOGY" and "DIABETES & ENDOCRINOLOGY".

Terms have been added to the manuscript submission site under Step 4: Attributes.

The best thing for me is the attempt to protocolise a way of carrying out a meta-analysis, although it does not seem to me to be a new contribution, I value it as an important aspect.

In short, it seems to me to be a good methodological study, with clinical discrepancies and I am not sure about the number of citations it can contribute to the journal.

Given the potential for publication or reporting biases, registration and publication of a pre-specified review protocol is recommended.⁵ This will enable readers of the final review to better determine if there is any deviation from the study plan, reducing the possibility of bias and preventing duplication of effort by other groups.

VERSION 2 – REVIEW

REVIEWER	Richard Shaw Institute of Translational Medicine, University of Liverpool,
REVIEW RETURNED	28-Mar-2022
GENERAL COMMENTS	Minor amendments addressed. The authors have listed very clearly their intentions and methodology.
REVIEWER	Francesc Xavier Aviles-Jurado Hospital Clinic de Barcelona, Otorhinolaryngology
REVIEW RETURNED	31-Mar-2022
GENERAL COMMENTS	All my objections have been resolved. Thank you very much